# Bedside Renal Doppler Ultrasonography and Acute Kidney Injury after TAVR

**DOI:** 10.3390/jcm9040905

**Published:** 2020-03-25

**Authors:** Marilou Peillex, Benjamin Marchandot, Sophie Bayer, Eric Prinz, Kensuke Matsushita, Adrien Carmona, Joe Heger, Antonin Trimaille, Hélène Petit-Eisenmann, Laurence Jesel, Patrick Ohlmann, Olivier Morel

**Affiliations:** 1Université de Strasbourg, Pôle d’Activité Médico-Chirurgicale Cardio-Vasculaire, Nouvel Hôpital Civil, Centre Hospitalier Universitaire, 67000 Strasbourg, France; mariloupeillex@gmail.com (M.P.); benjaminmarchandot@gmail.com (B.M.); matsuken_22@yahoo.co.jp (K.M.); adrien.carmona@chru-strasbourg.fr (A.C.); joe.heger@chru-strasbourg.fr (J.H.); antonin.trimaille@laposte.net (A.T.); helene.petit-eisenmann@chru-strasbourg.fr (H.P.-E.); laurence.jesel@chru-strasbourg.fr (L.J.); patrick.ohlmann@chru-strasbourg.fr (P.O.); 2Université de Strasbourg, Pôle de Biologie, Nouvel Hôpital Civil, Centre Hospitalier Universitaire, 67000 Strasbourg, France; sophie.bayer@chru-strasbourg.fr; 3Université de Strasbourg, Département de néphrologie et dialyse, Nouvel Hôpital Civil, Centre Hospitalier Universitaire, 67000 Strasbourg, France; eric.prinz@chru-strasbourg.fr; 4UMR CNRS 7213 Laboratoire de Biophotonique et Pharmacologie, Faculté de Pharmacie, Université de Strasbourg, 67400 Illkirch, France

**Keywords:** aortic stenosis, transcatheter aortic valve replacement, acute kidney injury, doppler based renal resistive index

## Abstract

Acute kidney injury (AKI) following transcatheter aortic valve replacement (TAVR) is associated with a dismal prognosis. Elevated renal resistive index (RRI), through renal Doppler ultrasound (RDU) evaluation, has been associated with AKI development and increased systemic arterial stiffness. Our pilot study aimed to investigate the performance of Doppler based RRI to predict AKI and outcomes in TAVR patients. From May 2018 to May 2019, 100 patients with severe aortic stenosis were prospectively enrolled for TAVR and concomitant RDU evaluation at our institution (Nouvel Hôpital Civil, Strasbourg University, France). AKI by serum Creatinine (sCr-AKI) was defined according to the VARC-2 definition and AKI by serum Cystatin C (sCyC-AKI) was defined as an sCyC increase of greater than 15% with baseline value. Concomitant RRI measurements as well as renal and systemic hemodynamic parameters were recorded before, one day, and three days after TAVR. It was found that 10% of patients presented with AKI_sCr_ and AKI_sCyC_. The whole cohort showed higher baseline RRI values (0.76 ± 0.7) compared to normal known and accepted values. AKI_sCyC_ had significant higher post-procedural RRI one day (Day 1) after TAVR (0.83 ± 0.1 vs. 0.77 ± 0.6, CI 95%, *p* = 0.005). AUC for AKI_sCyC_ was 0.766 and a RRI cut-off value of ≥ 0.795 had the most optimal sensitivity/specificity (80/62%) combination. By univariate Cox analysis, Mehran Risk Score, higher baseline right atrial pressure at baseline >0.8 RRI values one day after TAVR (HR 6.5 (95% CI 1.3–32.9; *p* = 0.021) but not RRI at baseline were significant predictors of AKI_sCyC_. Importantly, no significant impact of baseline biological parameters, renal or systemic parameters could be demonstrated. Doppler-based RRI can be helpful for the non-invasive assessment of AKI development after TAVR.


**New & Highlights**



*What is known about this topic?*
Acute kidney injury (AKI) is a common complication following Transcatheter Aortic Valve Replacement (TAVR)AKI occurs in a sizeable proportion of TAVR patients (22.1% ± 11.2 according to the VARC-2 definition) and carries poor prognosisIntrarenal Doppler ultrasonography can assess intrarenal hemodynamicDoppler based renal resistive index (RRI) measurements is a rapid and non-invasive method proposed for early AKI detection



*What does the paper add?*
24 h post-TAVR evaluation by Doppler-based resistive index is associated with AKI occurrence up to day 3Doppler based renal resistive index is an easy, objective, reliable and low-cost tool that succeeded to identify an at-risk population for AKI and able to improve the post TAVR managementThis study clarifies the characteristics of intrarenal Doppler RRI profiles and their interactions with systemic hemodynamic in TAVR patients


## 1. Introduction

Acute kidney injury (AKI) is a common complication following transcatheter aortic valve replacement (TAVR) and remains associated with a dismal prognosis. Based on the current Valve Academic Research Consortium (VARC)-2 definition [1], the reported incidence of post TAVR AKI is 22.1% ± 11.2 [2]. Depicting the scope of AKI in the field of TAVR relies on a variety of factors from impaired baseline renal function, hemodynamic instability during pacing, and use of contrast medium, to post procedural complications such as bleeding. Given that TAVR is expected to be increasingly performed, screening high-risk patients for AKI is of paramount importance since easy, accessible, and preventive measures, such as optimal periprocedural hydration and careful contrast use, are available. As AKI after TAVR is common and both transient and persistent AKI have been independently associated with higher mortality rates [3,4]. Thus, the prediction of AKI after TAVR and AKI prevention are key approaches in current practice.

Intrarenal Doppler ultrasonography is a well-recognized and non-invasive method aimed to better assess intrarenal hemodynamic. Renal Doppler ultrasound (RDU) can easily measure the renal resistive index (RRI), a sonographic index reflecting renal arterial blood flow profile, arterial stiffness, sensitivity to arterial resistance, and capacitance changes. Elevated RRI has been associated to adverse outcomes in hypertensive, diabetic, and elderly patients [5,6,7,8], increased mortality in chronic kidney disease [9], post procedural AKI development [10,11,12], and increased systemic arterial stiffness [13]. 

To date, there are few data regarding cardio-renal hemodynamics in the field of TAVR. Therefore, the aim of this pilot study was to (i) investigate the association of Doppler based RRI with acute kidney injury after the TAVR procedure and (ii) evaluate the dynamic change of various cardio-renal hemodynamic parameters according to AKI occurrence.

## 2. Methods

### 2.1. Patients

100 patients with severe aortic stenosis (AS) and high or intermediate surgical risk according to Logistic EuroSCORE were prospectively enrolled for TAVR and concomitant RDU evaluation at our institution (Nouvel Hôpital Civil, Strasbourg University, France) from May 2018 to May 2019. The study was approved by the local bioethics committee. All participants gave their informed written consent and agreed to the anonymous processing of their data. Commercially available valves, such as the balloon expandable Edwards SAPIEN XT or S3 prosthesis (Edwards Life sciences LLC, Irvine, CA, USA) and the self-expandable CoreValve or Evolute-R (Medtronic CV, Irvine, CA, USA), were used. Anesthesiological management included a local anaesthesia plus sedation. It consisted of lidocaine injected subcutaneously at the arterial and venous access sites (maximum dose 4 mg/kg), with sedation accomplished with remifentanil infusion adjusted according to the patient’s response (target level: score 2–3 with modified Wilson sedation scale; starting dose 0.025 µg/kg/min, maximum dose 0.2 µg/kg/min).

### 2.2. Definition of AKI 

Acute kidney injury by serum Creatinine (AKI_sCr_) was defined according to the VARC-2 definition (1) as an absolute increase in serum creatinine of ≥ 0.3 mg/dL (≥ 26.4 µmol/L) OR ≥ 50% increase in serum creatinine 72 h after TAVR. The AKIN Classification classified biochemical severity of AKI as follow: (i) Stage 1: Increase in serum creatinine to 150–199% (1.5–1.99× increase compared with baseline) (ii) Stage 2: Increase in serum creatinine to 200–299% (2.0–2.99× increase compared with baseline) (iii) Stage 3: Increase in serum creatinine to ≥ 300% (> 3× increase compared with baseline).

AKI as determined by Serum cystatin C was identified as AKI_sCyC_ and defined as an sCyC increase of greater than 15% with baseline value [14].

We sought to evaluate the incremental value of a new definition of AKI: AKI_sCr_ OR _sCyC_ detected by a single marker (sCr increase ≥ 0.3 mg/dL or 50% from baseline OR sCyC increase ≥ 15% from baseline) and AKI_sCr_ AND _sCyC_ where AKI is detected using by both markers: sCr increase ≥ 0.3 mg/dL or 50% from baseline and sCyC increase ≥ 15% from baseline. Patients were excluded if they were receiving dialysis at baseline.

### 2.3. Intrarenal Hemodynamic Evaluation by Doppler Ultrasonography

Patients with previous known renal artery disease (renal artery stenosis, occlusion or renal artery/vein thrombosis) were excluded from further analysis. RDU measurements were performed 12 h prior, 24 h and 72 h after TAVR procedures. According to previous work and validated methodology, renal resistive index (RRI) was measured with an ultrasound-Doppler pulsed-wave Doppler probe (5S ultrasound transducer, General Electric Medical Systems) on a Vivid S7 ultrasound system by two independent trained physicians. After locating the kidneys, intrarenal renal vessels were identified using color Doppler and sampling for RRI screened at the level of interlobar arteries. Measurements using pulse-wave Doppler were repeated in different parts (superior, median, and lower) of the kidney and at least three reproducible and consecutive waveforms were obtained to measure RRI parameters (Figure 1). The RRI index was calculated according to the following formula: ((peak systolic velocity − end diastolic velocity)/peak systolic velocity). The mean value of three different measurements was recorded. The mean reference value for normal RRI in adults is 0.60 ± 0.10 and 0.70 considered as the upper limit of normal [15,16,17]. 

### 2.4. Echocardiography Protocol and Hemodynamic Parameters

Concomitant echocardiography was performed after RDU measurements 12 h prior, 24 h and 72 h after TAVR procedure. The following echocardiographic and clinical variables were collected from each patient: heart rate; systolic, diastolic, and mean blood pressures; left ventricular ejection fraction (LVEF); aortic flow time-velocity integral; Aortic velocity time integral and diastolic left ventricular function parameters. Systemic and local renal hemodynamic parameters were calculated according to current definitions (Appendix A) and included valvulo-arterial impedance, total arterial load, pulse pressure, systemic arterial compliance, resistive arterial load, renal arterial load, renal pulse pressure, and renal arterial compliance.

### 2.5. Collection of Data and Outcomes

All baseline and follow-up variables were recorded and entered into a secure, ethics-approved database. Clinical endpoint including acute kidney injury, mortality, stroke, bleeding, access-related complications and conduction disturbances were assessed according to the definitions provided by the Valve Academic Research Consortium-2 (VARC-2). All clinical events were adjudicated by an events validation committee.

The primary endpoint of the study was the AKI incidence 72 h after TAVR. The secondary endpoint was a composite endpoint defined by mortality, stroke, and hospitalization for heart failure. All patients were contacted by phone and questioned by a standardized questionnaire about their health status, symptoms, medications and the occurrence of adverse events.

### 2.6. Statistical analysis

Quantitative variables were described according to AKI occurrence and expressed as means ± standard deviation. Categorical variables were expressed as counts and percentages. Categorical variables were compared with chi-square tests or Fisher’s exact tests. Continuous variables were compared with the use of parametric (ANOVA) or non-parametric Mann–Whitney tests as appropriate. Normality of the distribution was tested using Kolmogorov—Smirnov Test. To determine predictors of AKI, regression analysis was performed. A *p* value < 0.05 was considered significant. Receiver-operating characteristic (ROC) curve analysis was performed to establish the threshold values most predictive of AKI. Calculations were performed using SPSS 17.0 for Windows (SPSS Inc., Chicago, IL, USA).

## 3. Results

### 3.1. Baseline Characteristics 

Baseline, procedural, and biological characteristics are summarized in Table 1, Table 2 and Table 3. Of the 100 TAVR patients recruited, AKI was documented respectively for 10%; 10%, 16%, and 4% of the global cohort according to the AKI_sCr_, AKI_sCyC_, AKI_sCr,_ OR _sCyC_ and AKI_sCr_ AND _sCyC_ definitions respectively (Table 4). Regarding AKI as determined by serum creatinine (AKI_sCr_), stage 1 AKI according to the AKIN system occurred in eight patients, stage 2 in two patients, and none in stage 3. No difference regarding traditional cardiovascular risk factors according to previous medical history apart from coronary artery disease could be evidenced between groups. Of note, AKI_sCyC_ patients showed a higher Mehran contrast-induced nephropathy risk score (*p* = 0.016) while no difference regarding contrast media volume administration or length of procedure. No pre-operative complications could be evidence such as cardiac arrest, tamponade, immediate red blood cell transfusion nor vasopressor/inotropic support requirement. Likewise, no patients required dialysis therapy nor the short-term use of mechanical circulatory support devices.

### 3.2. Renal Resistive Index (RRI)

The whole cohort of AS patients eligible for TAVR showed higher baseline RRI values (0.76 ± 0.7) compared to normal known and accepted values. While RRI at baseline and Day 3 were similar between groups, patients with AKI_sCyC_ had significant higher post-procedural RRI one day (Day 1) after TAVR (Table 5). Receiver-operating characteristic curve was applied for the identification of an optimal RRI value (Figure 2). AUC for AKI_sCyC_ was 0.766 and a RRI cut-off value of ≥ 0.795 had the most optimal sensitivity/specificity (80/62%) combination. Similarly, higher post-procedural RRI one day after TAVR could be evidenced according to the AKI_sCr_ OR _sCyC_ and AKI_sCr_ AND _sCyC_ definition (Table 6).

### 3.3. Hemodynamic Parameters 

Likewise, AKI_sCyC_ patients showed transient higher valvulo-arterial impedance and total arterial load one day after TAVR (Table 7). Conversely and interestingly, no difference regarding inotropy (cardiac performance assessed by stroke volume and cardiac index), left ventricular filling pressure (E/A, E/e′) or renal hemodynamic parameter could be established. A higher preload at baseline evidenced by higher right atrial pressure (echocardiographic assessment of estimated right atrial pressure) was associated with AKI_sCyC_ patients.

### 3.4. Predictors of AKI Following TAVR

By univariate Cox analysis, Mehran Risk Score, higher baseline right atrial pressure at baseline, > 0.8 RRI values one day after TAVR (HR 6.5 (95% CI 1.3–32.9; *p* = 0.021) but not RRI at baseline were significant predictors of AKI_sCyC_ (Table 8). 

Importantly, no significant impact of baseline biological parameters, renal or systemic parameters could be demonstrated. 

## 4. Discussion

AKI is a common complication in patients with TAVR and associated with adverse clinical outcomes [2]. Although prevalence and preventive measures for AKI in TAVR have been investigated in multiple clinical trials [18,19], methods designed to accurately detect early AKI are still lacking. Therefore, new diagnostic tools that could predict AKI and be commonly adopted by cardiologists are desperately needed. To the best of our knowledge, this is the first study evaluating the use and role of intrarenal Doppler ultrasonography in diagnosing AKI in an unselected TAVR population. 

The results of our study show that, in AS patients undergoing TAVR, AKI patients exhibited (i) higher RRI values compared to normal known and accepted values; (ii) higher transient and residual afterload assessed by valvulo-arterial impedance and total arterial load one day after TAVR and finally (iii) no difference regarding renal hemodynamic parameter.

### 4.1. Impact of AKI Definitions

The main purposes of this study were to assess (i) the optimal RRI cutoff point to detect AKI and (ii) test whether Serum cystatin C and combined serum cystatin C and Serum cystatin C definition for AKI diagnosis provides extra benefit. sCyC has been validated as an alternative to sCr for diagnosing AKI [20]. Likewise, sCyC has been proposed as a more sensitive AKI marker [21,22,23]. Of importance, Cystatin C is not affected by muscle mass or diet and is less strongly associated with age, sex, and race than creatinine [24] and makes this biomarker of particular interest in AS patients and particularly among the TAVR population. 

### 4.2. Impact of Renal Resistive Index (RRI)

The study’s findings highlight that higher immediate post-procedural RRI is strongly associated with AKI. The present observation is of clinical importance as RRI is a rapid, non-invasive and repeatable strategy that may help in assessing renal preclinical dysfunction and in evaluating subsequent risk of acute kidney injury. Additionally, as RRI can be performed in most patients by inexperienced operators following a half-day course [25] our data clearly challenge the routine and widespread use of bedside renal Doppler ultrasonography in the post TAVR management. Though, specific technical and clinical nuances must be raised in the field of TAVR patients. 

First, due to arterial stiffness, TAVR patients exhibited higher baseline RRI values (0.76 ± 0.7) compared to normal known and accepted values: 0.60 ± 0.10 in adults and 0.70 considered as the upper limit of normal [15,16,17]. Indeed, RRI has been reported to increase in the healthy elderly population [26,27,28,29] and age-related changes in vascular compliance: two intrinsic features defining the TAVR population. Common RRI cut-off values are presently beyond the field of elderly AS patient’s eligible for TAVR, but RRI > 0.795 (Se 80%; 1-Sp 0.378) provided interesting diagnostic performance. The clinical significance of the extended proposed threshold and its diagnostic performance remains to be validated in adequately powered studies. 

Second, RRI is the result of a complex interaction between intrarenal circulation and systemic hemodynamics. Not only is RRI a specific marker of kidney damage, but it shares strong relationships with the systemic circulation and some cardiovascular parameters such as cardiac function, aortic stiffness, vascular compliance, renal capillary wedge pressure, and clinical settings such as heart failure progression and regression [30]. We might have just performed a futile exercise in the field of RRI and cardio-renal interactions after TAVR in resuming RRI to either renal, systemic hemodynamics, heart failure or cardiorenal syndrome relief. We sought to specifically explore the hemodynamic factors in play that should be considered to understand RRI clinical meaning and to explain how systemic and renal parameters both can affect the RRI analysis as TAVR represents a unique therapeutic modality in acute hemodynamic changes. All this without forgetting that RRI goes beyond considering only one specific feature (renal, hemodynamics, renal capillary wedge pressure etc.) of its complex and dynamic definition. 

### 4.3. Impact of Systemic and Renal Hemodynamics Parameters

The mechanistic model of acute relief of the valvular load provided by TAVR procedures enables the measurement and investigation of rapid systemic and renal hemodynamic changes. Albeit limited to a small sample size, we provided data suggesting that higher RRI is linked to transient higher valvulo-arterial impedance (Zva) and total arterial load (TAL) one day after TAVR. 

In patients with AS, high Zva and TAL are accurate markers of advanced disease and have been previously associated with adverse outcomes both in AS [31,32] and post TAVR patients [33,34]. Our report stands out as the first insight demonstrating that transient residual high post TAVR afterload impacts AKI development through higher RRI, a surrogate marker of renal arterial resistance.

We believe the results of our findings suggest that systemic arterial afterload and RRI evaluation should be incorporated to optimize patient periprocedural TAVR management. Long-term prospective studies are needed to establish the prognostic value of measuring such indices before and after valve interventions and how such indices may aid selecting the best management strategy for patients after TAVR.

Our findings reinforce the understanding of the complex nature of RRI changeability and renal function in the field of TAVR with higher RRI depending on (i) fixed systemic and renal vascular resistance despite acute relief from AS provided by TAVR; (ii) absence of compensatory mechanism maintaining or improving GFR such as relief from heart failure and inherent cardiorenal syndrome and (iii) on the other hand true renal damage induced by contrast media, hypovolemia etc.

Our experimental data suggest therefore that Doppler-based RRI could be an integrative parameter that may help in detecting early renal dysfunction, portray the post procedural systemic vascular load and become a surrogate maker of cardiorenal interaction after TAVR.

Finally, RRI at 24 h after TAVR was not influenced by preoperative renal function but hemodynamic change, and it was thus considered as a surrogated marker. These results are important to assess the mechanism of AKI after TAVR. Indeed, due to the small number of subjects included and the complex pathophysiological factors as displayed in this paper, we could not establish that RRI evaluated preoperatively could predict AKI.

### 4.4. Study Limitations

Several limitations should be taken into account in the interpretation of the data. First, the observational design of the study, though prospective inclusions, did not allow us to establish a cause-effect relationship between the observed associations. Additionally, the inclusion of overwhelming elderly patients with preserved LVEF and moderately altered GFR (the mean LVEF of patients in our study was > 55% and mean sCr-GRF 54.5 µmol/L) limited the generalizability of the results to other populations. Indeed, further data supporting the link between altered cardiac output, renal function, and RRI evaluation (e.g., low-flow low-gradient AS or reduced LVEF patients) may represent a population of choice to study due to prominent hemodynamic changes and cardiorenal interactions after TAVR. While AKI is widely known to be associated with red blood cell transfusions, such association was not observed given the limited size of the cohort. Second, international standardization of the Cystatin C assay is not finalized. Third intra and Inter-observer RRI variability were not analysed but assumed to be very low in the literature. Fourth, concomitant therapeutic modalities after TAVR known to affect RRI (e.g., diuretic therapy with the consequent reduction in renal venous pressure) were not recorded. Finally, we conducted a pilot study, with a consequently small sample size (100 patients) and only 10 events, making any analysis, especially a logistic regression, tentative and for the purpose of hypothesis generation. The AUC of 0.766 should be considered with caution and interpreted in the light of a low number of events. 

## 5. Conclusions

Doppler-based renal resistive index can be helpful for the non-invasive assessment of acute kidney injury development after transcatheter aortic valve replacement.

## Figures and Tables

**Figure 1 jcm-09-00905-f001:**
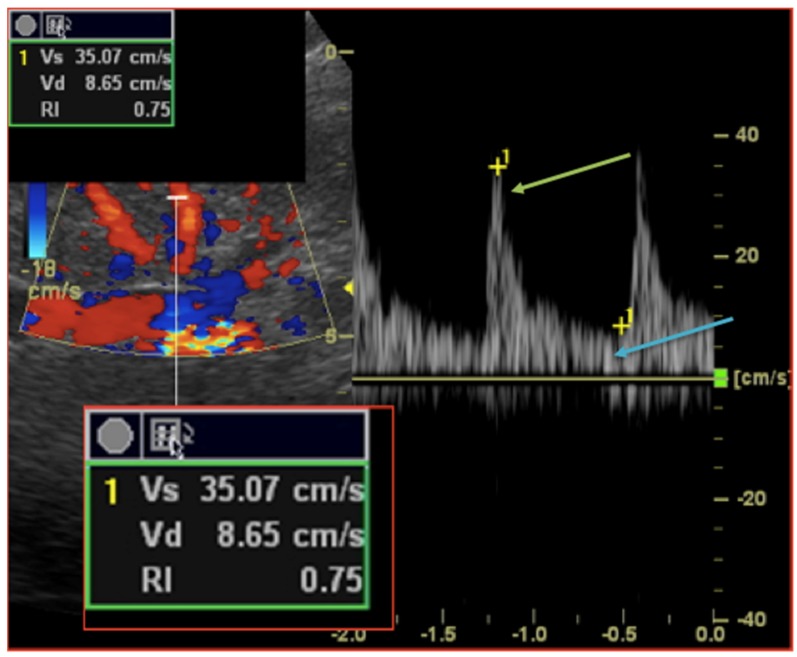
Intrarenal Doppler Ultrasonography: Renal Resistive index measurement technique. A sample volume is placed within an interlobar artery using Colour Doppler guidance. Spectral analysis of vascular signals is then obtained, and measurement callipers are set as follow: (i) systolic peak (green arrow); (ii) end diastole peak (blue arrow). Renal Resistive index is then calculated according to the formula (Vs − Vs)/Vs.

**Figure 2 jcm-09-00905-f002:**
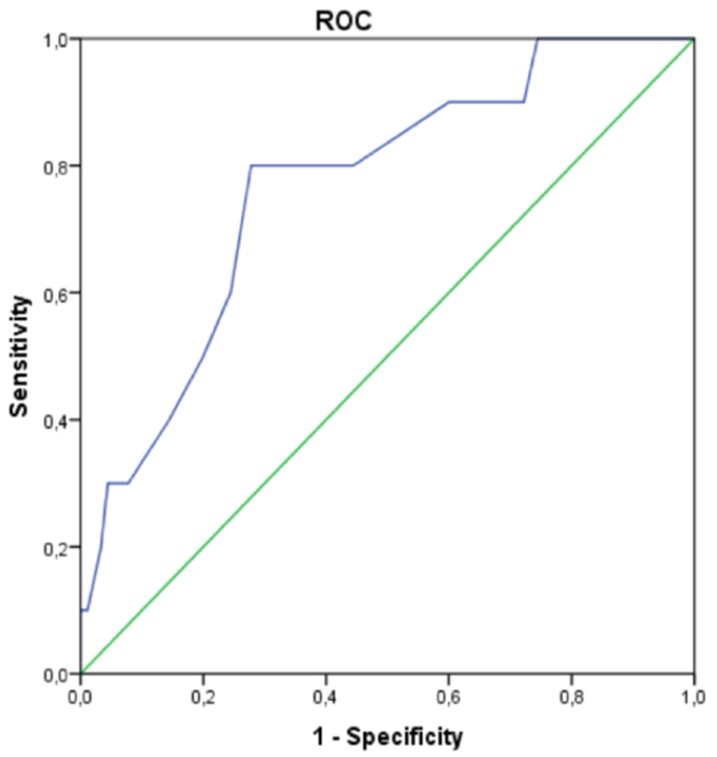
Receiver-Operating Characteristic (ROC) Curve Displaying the Optimal Threshold and Goodness of Renal Resistive Index one day after TAVR for sCyC AKI. Renal Resistive Index one day after TAVR provides a good prediction of sCyC AKI with an optimal cutoff value of 0.795 and an area under the curve of 0.766. AUC = 0.766; Confidence interval (0.618; 0.913, *p* = 0.006); Optimal RRI threshold = 0.795 (Se 80%; 1-Sp 0.378).

**Table 1 jcm-09-00905-t001:** Baseline characteristics.

	Global Cohort	AKI sCyC	No AKI sCyC	*p*-Value
	*n* = 100	*n* = 10	*n* = 90	
**Clinical parameters**				
Age—years ± SD	83.7 ± 6.3	83.4 ± 6.7	83.7 ± 6.3	0.88
Male sex—no/total no. (%)	48 (48%)	5 (50%)	43 (48%)	0.58
Euroscore * (%)	5.5 ± 5	8 ± 8	5.3 ± 4.6	0.1
STS mortality (%)	5.2 ± 3.7	6.7 ± 4.4	5 ± 3.6	0.16
STS renal failure (%)	6.5 ± 8.5	10.2 ± 10.9	6.1 ± 8.2	0.16
Mehran Contrast nephropathy risk score (points)	7.75 ± 3	9.4 ± 4	7.6 ± 2.8	0.067
Mehran Risk Score (%)	13.9 ± 7.6	19.4 ± 14.7	13.3 ± 6.3	0.016
Coronary artery disease—no./total no. (%)	53 (53%)	2 (20%)	51 (56.7%)	0.03
Pacemaker—no./total no. (%)	12 (12%)	1 (10%)	11 (12.2%)	0.66
**Cardiovascular Risk Factors**				
Hypertension—no./total no. (%)	90 (90%)	9 (90%)	81 (90%)	0.67
Past or current smoker—no./total no. (%)	26 (26%)	4 (40%)	22 (24.4%)	0.24
Dyslipidaemia—no./total no. (%)	53 (53%)	5 (10%)	48 (52%)	0.55
Diabetes mellitus—no./total no. (%)	40 (40%)	5 (10%)	35 (38.9%)	0.36
BMI—kg/m^2^ ± SD	32 ± 13	29 ± 9	32 ± 13	0.47
**Prehospital management**				
VKA—no./total no. (%)	18 (18%)	1 (10%)	17 (18.9%)	0.42
DOAC—no./total no. (%)	22 (22%)	3 (30%)	19 (21.1%)	0.38
ASA—no./total no. (%)	56 (56%)	4 (5.6%)	52 (57.8%)	0.23
Clopidogrel—no./total no. (%)	23 (23%)	0	23 (25.6%)	0.06
ACE inhibitors/ARBs—no./total no. (%)	56 (56%)	5 (50%)	51 (56.7%)	0.47
Beta blockers—no./total no. (%)	48 (48%)	5 (50%)	43 (47.8%)	0.58
Calcium channel blockers—no./total no. (%)	31 (31%)	2 (20%)	29 (32.2%)	0.35
Thiazide diuretics—no./total no. (%)	16 (16%)	3 (30%)	13 (14.4%)	0.19
Aldosterone-receptor antagonists (ARAs)—no./total no. (%)	12 (12%)	0	12 (13.5%)	0.26
Furosemide—mg ± SD	71 ± 147	84 ± 155	70 ± 147	0.77
Statin—no./total no. (%)	46 (46%)	3 (30%)	43 (47.8%)	0.23
**Echocardiography**				
LEVF—% ± median IQR	60 (51–67)	54.6 ± 12	59 ± 12.4	0.19
LV mass—g/m^2^ ± SD	137 ± 77.8	112.7 ± 19.9	138.8 ± 80.8	0.43
LVendDV—mm ± SD	49 ± 8.5	46.2 ± 9	49 ± 8.4	0.33
Mean Aortic Gradient—mmHg ± SD	44.5 ± 11.9	43.5 ± 9.8	44.6 ± 12.2	0.72
E/A	0.9 ± 0.6	0.6 ± 0.2	1 ± 0.6	0.95
E/e′	11.9 ± 4.6	12.9 ± 3.9	11.7 ± 4.6	0.48
Mean Pulmonary Artery Pressure (MPAP)—mmHg ± SD	40.5 ± 13.2	42 ± 11.1	40.3 ± 13.4	0.77
Right Atrial Pressure (RAP)—mmHg ± SD	7 ± 4	10 ± 6	6 ± 4	0.02
Stroke volume (SV)—mL ± median (IQR)	81.5 (65.3–99.5)	88.1 ± 22.2	80 (65.7–98.5)	0.34
Cardiac index—mL/min/m^2^ ± SD	2.9 ± 0.9	3.3 ± 1.3	2.8 ± 0.9	0.17
**Baseline biological parameters**				
Creatinine (Cr) level—µmol/L ± SD	113.6 ± 77.9	135.9 ± 89.9	111.2 ± 76.6	0.34
Cr eGFR—mL/min/1.73 m^2^ ± SD	54.5 ± 19.9	49 ± 23.5	55 ± 19.6	0.37
Cystatin (CysC)—mg/L ± SD	1.7 ± 0.9	1.9 ± 1	1.7 ± 0.9	0.59
CysC eGFR—ml/min/1.73m^2^ ± SD	43.9 ± 18.3	38.5 ± 18.7	44.5 ± 18.3	0.32
Haemoglobin—g/dL median (IQR)	12 (11–13.1)	11.1 ± 2.1	12 ± 2	0.17
BNP—ng/L ± SD	471 ± 856	561 ± 848	461 ± 862	0.73

Data are presented as mean ± or *n* (%) or median (25th–75th percentile). ACE inhibitor = Angiotensin-converting enzyme inhibitor; AKI = Acute Kidney Injury; ARBs = Angiotensin II receptor blockers; ASA = Aspirin; BMI = body mass index; BNP = B-type natriuretic peptide; Cr = creatinine; CysC = Cystatin C; DOAC = direct oral anticoagulant; EuroSCORE = logistic EuroSCORE predicted risk of mortality at 30 days; GFR = glomerular filtration rate; LV = left ventricle; LVEF = left ventricular ejection fraction; sCr = serum creatinine; sCyC = serum Cystatin C; STS score = Society of Thoracic Surgeon; VKA = Vitamin K antagonists. * The logistic European System for Cardiac Operative Risk Evaluation (EuroSCORE) is calculated by means of a logistic-regression equation; online and downloadable versions of the EuroSCORE calculator are available on the EuroSCORE Web site.

**Table 2 jcm-09-00905-t002:** Procedural characteristics.

	Global Cohort	AKI sCyC	No AKI sCyC	*p*-Value
	*n* = 100	*n* = 10	*n* = 90	
**Approach**				
Transfemoral—no./total no. (%)	93 (93%)	9 (90%)	84 (93%)	0.48
Transcarotid—no./total no. (%)	5 (5%)	0	5 (5.6%)	0.58
Transaortic—no./total no. (%)	1 (1%)	1 (10%)	0	0.1
**Valve**				
Sapien—no./total no. (%)	64 (64%)	7 (70%)	57 (63.3%)	0.48
Corevalve—no./total no. (%)	34 (34%)	3 (30%)	31 (34.4%)	0.54
Boston Accurate—no./total no. (%)	2 (2%)	0	2 (2.2%)	0.81
Sizing—no./total no. (%)				
23 mm	18 (18%)	2 (20%)	16 (17.8%)	0.57
25 mm	1 (1%)	0	1 (1.1%)	0.9
26 mm	40 (40%)	4 (40%)	36 (40%)	0.64
27 mm	1 (1%)	0	1 (1.1%)	0.9
29 mm	1 (1%)	0	1 (1.1%)	0.9
31 mm	33 (33%)	4 (40%)	29 (32.2%)	0.43
34 mm	6 (6%)	0	6 (6.7%)	0.52
Post Dilatation—no./total no. (%)	6 (6%)	0	6 (6.7%)	0.52
**Procedure**				
Length of the procedure—min ± DS	69 ± 21	78 ± 30	68 ± 20	0.16
Contrast media volume—mL ± DS	140 ± 50	138 ± 47	141 ± 50	0.88
**Procedural and Post-procedural Complications**				
Major vascular complications—*n* (%)	9 (9%)	0	9 (10%)	0.37
Minor vascular complications—*n* (%)	23 (23%)	2 (20%)	21 (23.3%)	0.58
Red blood cell transfusion—*n* (%)	0.5 ± 1	1 ± 1.7	0.4 ± 0.9	0.08
Length of Stay (days)	8.4 ± 4.6	8.4 ± 3.7	8.4 ± 4.7	0.98

Data are presented as mean ± or *n* (%). AKI = Acute Kidney Injury; BNP = B-type natriuretic peptide; ScyC = serum Cystatin C.

**Table 3 jcm-09-00905-t003:** Biological parameters.

	Global Cohort	AKI SCyC	No AKI SCyC	*p*-Value
	*n* = 100	*n* = 10	*n* = 90	
**Serum Creatinine level—µmol/L**				
Baseline	113.6 ± 77.9	135.9 ± 89.9	111.2 ± 76.6	0.34
Post TAVR—H_0_	103.2 ± 98.8	141.7 ± 150.5	98.8 ± 91.4	0.2
Post TAVR—Day 1	108.6 ± 115.7	172.6 ± 193.5	101.5 ± 103	0.065
Post TAVR—Day 3	108.5 ± 95.5	177.1 ± 174.4	100.9 ± 80.4	0.016
Serum Cystatin—mg/L				
Baseline	1.7 ± 0.9	1.9 ± 1	1.7 ± 0.9	0.59
Post TAVR—Day 1	1.6 ± 1	2.2 ± 1.6	1.5 ± 0.8	0.025
Post TAVR—Day 3	1.6 ± 1	2.3 ± 1.7	1.6 ± 0.8	0.029
Haemoglobin—g/dL				
Baseline	12 (11–13.1)	10.2 (9.3–10.9)	11.2 (10.2–12.2)	0.17
Post TAVR—Day 1	10.2 (9.6–10.7)	10.2 (9.6–10.7)	10.6 (9.5–11.6)	0.25
Post TAVR—Day 3	9.6 (8.5–9.95)	9.6 (8.5–9.9)	10 (9.2–11)	0.08
BNP—ng/L ± SD				
Baseline	471 ± 856	561 ± 848	461 ± 862	0.73
Post TAVR—H_0_	515 ± 920	690 ± 1140	495 ± 898	0.53
Post TAVR—Day 1	426 ± 747	686 ± 1185	397 ± 684	0.24
Post TAVR—Day 3	283 ± 402	511 ± 674	257 ± 356	0.058
Post TAVR—Day 3	60.8 ± 47	66.2 ± 86.7	60.2 ± 41	0.7

Data are presented as mean ± or *n* (%) or median (25th–75th percentile). AKI = Acute Kidney Injury; TAVR = Transcatheter aortic valve replacement.

**Table 4 jcm-09-00905-t004:** Acute Kidney injury (AKI) definitions and incidence.

Group	Definition	no./Total no. (%)
Group 1: No AKI	No AKI: No AKIsCr AND No AKI sCyC	(84/100)—84%
Group 2: AKIsCr	AKI sCr: according to VARC2 definition. Absolute increase in sCr of ≥ 0.3 mg/dL (≥ 26.4 mmol/L) OR ≥ 50% increase in sCr	(10/100)—10%
Group 3: AKIsCyC	AKI sCyC: sCyC increase ≥ 15% from baseline.	(10/100)—10%
Group 4: AKI sCr OR sCyC	AKI sCr OR sCyC AKI detected by a single marker: fulfill only 1 of criteria as below: (1) sCr increase ≥ 0.3 mg/dL or 50% from baseline OR (2) sCyC increase ≥ 15% from baseline.	(16/100)—16%
Group 5: AKI sCr AND sCyC	AKI sCr AND sCyC: AKI detected by both markers: sCr increase ≥ 0.3 mg/dL or 50% from baseline; and sCyC increase ≥ 15% from baseline.	(4/100)—4%

Data are presented as mean ± or *n* (%). AKI = Acute Kidney Injury; sCr = serum creatinine; sCyC = serum Cystatin C; VARC2 = valve academic research consortium-2 consensus.

**Table 5 jcm-09-00905-t005:** Doppler based resistive index and hemodynamic parameters.

	Global Cohort	AKI SCyC	No AKI SCyC	*p*-Value
	*n* = 100	*n* = 10	*n* = 90	
**Renal Doppler based parameters**				
Peak systolic velocity—cm/s ± SD				
Baseline	29.2 ± 9.5	28.9 ± 7.7	29.3 ± 9.7	0.91
Post TAVR—Day 1	32.5 ± 11	33.7 ± 6.9	32.4 ± 11.4	0.72
Post TAVR—Day 3	31.2 ± 8.7	26.7 ± 8.6	31.7 ± 8.6	0.083
End diastolic velocity—cm/s ± SD				
Baseline	6.8 ± 2.1	6.3 ± 1.4	6.8 ± 2.2	0.47
Post TAVR—Day 1	7 ± 3.3	5.7 ± 1.5	7.2 ± 3.4	0.17
Post TAVR—Day 3	6.9 ± 2.8	6.9 ± 4	6.9 ± 2.6	0.98
Renal doppler resistive index (RRI)				
Baseline	0.76 ± 0.7	0.78 ± 0.4	0.75 ± 0.7	0.34
Post TAVR—Day 1	0.78 ± 0.6	0.83 ± 0.1	0.77 ± 0.6	0.005
Post TAVR—Day 3	0.77 ± 0.6	0.75 ± 0.1	0.78 ± 0.5	0.11
RRI Day 1 > 0.7 (no./total no. (%))	90 (90%)	10 (100%)	80 (80%)	0.37
RRI Day 1 > 0.8 (no./total no. (%))	42 (42%)	8 (80%)	34 (37.8%)	0.013
RRI Day 3 > 0.7 (no./total no. (%))	93 (93%)	8 (80%)	85 (94.4%)	0.14
RRI Day 3 > 0.8 (no./total no. (%))	40 (40%)	4 (40%)	36 (40%)	0.64
**Echocardiography**				
**Mean Aortic Gradient—mmHg ± SD**				
Baseline	44.5 ± 11.9	43.5 ± 9.8	44.6 ± 12.2	0.72
Post TAVR—Day 1	8.4 ± 2.6	7.9 ± 3.5	8.5 ± 3.6	0.64
Post TAVR—Day 3	9.4 ± 4.2	9.9 ± 5	9.4 ± 4.1	0.71
**Stroke volume—mL ± median IQR**				
Baseline	81.5 (65.2–99.5)	92.5 (63–106.2)	80 (65.7–98.5)	0.34
Post TAVR—Day 1	75 (64.5–89.7)	78 (55–97)	75 (65.5–89)	0.87
Post TAVR—Day 3	78.5 (64.2–90.7)	84.5 (64.5–104.7)	77.5 (64–90.3)	0.5
**Cardiac index—mL/min/m^2^**				
Baseline	2.9 ± 0.9	3.3 ± 1.3	2.8 ± 0.9	0.17
Post TAVR—Day 1	2.9 ± 0.9	3 ± 0.8	2.9 ± 0.9	0.86
Post TAVR—Day 3	2.9 ± 0.98	3 ± 1.2	2.9 ± 0.96	0.59
**E/A**				
Baseline	0.9 ± 0.6	0.6 ± 0.2	1 ± 0.6	0.95
Post TAVR—Day 1	0.9 ± 0.5	0.95 ± 0.6	0.9 ± 0.5	0.85
Post TAVR—Day 3	1.4 ± 2.6	0.7 ± 0.1	1.5 ± 2.8	0.53
**E/e′**				
Baseline	11.9 ± 4.6	12.9 ± 3.9	11.7 ± 4.6	0.48
Post TAVR—Day 3	10.4 ± 3.8	11.9 ± 3.6	10.2 ± 3.8	0.21
Post TAVR—Day 3	11.2 ± 4.3	11.7 ± 1.8	11.1 ± 4.6	0.69
**Right Atrial Pressure—mmHg ± SD**				
Baseline	7 ± 4	10 ± 6	6 ± 4	0.02
Post TAVR—Day 1	6.5 ± 3	8 ± 4	6 ± 3	0.09
Post TAVR—Day 3	7 ± 4	9 ± 6	7 ± 4	0.31

Data are presented as mean ± or *n* (%) or median (25th–75th percentile). AKI = acute kidney injury; RAP = right atrial pressure; RRI = renal resistive index; SV = stroke volume; sCyC = serum Cystatin C; TAVR = Transcatheter aortic valve replacement.

**Table 6 jcm-09-00905-t006:** Renal Resistive index (RRI) values one-day after TAVR according to AKI definitions.

AKI Definition	RRI J1	*p*
Group 1: No AKI	0.78 ± 0.6	*x*
Group 2: AKI sCr	0.8 ± 0.05	0.149
Group 3: AKI sCyC	0.83 ± 0.04	0.05
Group 4: AKI sCr OR sCyC	0.81 ± 0.05	0.033
Group 5: AKI sCr AND sCyC	0.85 ± 0.05	0.013

Data are presented as mean ± or *n* (%). AKI = Acute kidney injury; RRI = renal resistive index; sCr: serum creatinine; sCyC = serum Cystatin C.

**Table 7 jcm-09-00905-t007:** Systemic and renal hemodynamic parameters.

	Global Cohort	AKI sCyC	NO AKI sCyC	*p*-Value
	*n* = 100	*n* = 10	*n* = 90	
**Renal parameters**				
Renal pulse pressure—mmHg ± SD				
Baseline	84 ± 14	87 ± 18	83 ± 14	0.38
Post TAVR—Day 1	82 ± 14	78 ± 14	82 ± 14	0.39
Post TAVR—Day 3	78 ± 14	82 ± 19	78 ± 13	0.36
Renal arterial load				
Baseline	10 ± 3.5	10.1 ± 4.4	10 ± 3.5	0.87
Post TAVR—Day 1	10 ± 3.3	9.8 ± 3.4	10 ± 3.3	0.76
Post TAVR—Day 3	9.2 ± 2.5	9.5 ± 3.7	9.2 ± 2.3	0.72
Renal arterial compliance				
Baseline	0.11 ± 0.4	0.1 ± 0.6	0.11 ± 0.4	0.27
Post TAVR—Day 1	0.11 ± 0.4	0.12 ± 0.5	0.11 ± 0.4	0.39
Post TAVR—Day 3	0.12 ± 0.4	0.12 ± 0.4	0.12 ± 0.4	0.77
**Systemic parameters**				
Valvuloarterial impedance—mmHg/mL/m^2^				
Baseline	4.3 ± 1.3	4.2 ± 1.5	4.3 ± 1.3	0.77
Post TAVR—Day 1	3.7 ± 1.6	4.9 ± 4	3.6 ± 1.1	0.016
Post TAVR—Day 3	3.2 ± 0.9	3.3 ± 1	3.2 ± 0.9	0.9
Total arterial load				
Baseline	2.9 ± 1	3 ± 1.2	2.9 ± 1	0.95
Post TAVR—Day 1	1.7 ± 0.9	2.4 ± 2.3	1.7 ± 0.5	0.008
Post TAVR—Day 3	1.5 ± 0.41	1.5 ± 0.5	1.5 ± 0.4	0.967
Systemic arterial compliance				
Baseline	0.9 ± 0.1	0.7 ± 0.3	1 ± 0.3	0.71
Post TAVR—Day 1	1.2 ± 4.3	1 ± 0.5	1.2 ± 0.4	0.23
Post TAVR—Day 3	0.7 ± 0.3	0.69 ± 0.2	0.7 ± 0.3	0.78
Pulse Pressure mmHg ± SD				
Baseline	65 ± 20	72 ± 12	64 ± 20	0.26
Post TAVR—Day 1	63 ± 20	70 ± 11	62.4 ± 21	0.3
Post TAVR—Day 3	63 ± 20	69 ± 10	62 ± 21	0.3
Resistive arterial load—dynes/s/cm—5 ± SD				
Baseline	2905 ± 1217	2743 ± 1272	2923 ± 1217	0.66
Post TAVR—Day 1	2755 ± 1506	3500 ± 3454	2672±1117	0.09
Post TAVR—Day 3	2631 ± 1069	2512 ± 888	2644±1091	0.71
Systolic Blood Pressure—mmHg ± SD				
Baseline	135 ± 22	143 ± 22	134 ± 22	0.21
Post TAVR—Day 1	132 ± 22	129 ± 22	132 ± 22	0.66
Post TAVR—Day 3	127 ± 25	133 ± 18	126 ± 25	0.42
Diastolic Blood Pressure—mmHg ± SD				
Baseline	70 ± 12	71 ± 17	70 ± 12	0.7
Post TAVR—Day 1	65 ± 13	63 ± 9	65 ± 13	0.69
Post TAVR—Day 3	64 ± 9	64 ± 10	64 ± 10	0.93
Mean Arterial Pressure—mmHg ± SD				
Baseline	91 ± 13	97 ± 16	90 ± 13	0.97
Post TAVR—Day 1	88 ± 13	86 ± 12	88 ± 13	0.59
Post TAVR—Day 3	86 ± 13	91 ± 17	85 ± 13	0.19

Data are presented as mean ± or *n* (%). AKI = acute kidney injury; TAVR = Transcatheter aortic valve replacement.

**Table 8 jcm-09-00905-t008:** Univariate COX regression for the development of AKI assessed by serum cystatine up to 3 days after TAVR.

	Univariate		
	HR	CI 95%	*p*-Value
**Baseline characteristics**			
Age	0.99	0.89–1.1	0.88
Male Sex	1	0.3–4	0.89
EuroScore	1	0.98–1.2	0.12
Mehran contrast nephropathy risk score (points)	1.2	0.98–1.5	0.074
Mehran Risk Score (%)	1.1	1–1.2	0.04
Hypertension	1	0.1–8.8	1
Current or past Smoking	2	0.5–8	0.29
Dyslipidaemia	0.88	0.24–3.2	0.84
Diabetes melitus	1.6	0.4–5.8	0.5
BMI	1	0.94–1.1	0.47
Coronary Artery Disease (CAD)	0.19	0.04–0.95	0.04
ACE inhibitors/ARBs	0.77	0.21–2.8	0.69
Furosemide	1	0.99–1	0.77
**Procedural characteristics**			
Length of procedure	1	0.99–1.1	0.16
Contrast media volume	0.99	0.99–1	0.88
Transfemoral approach	0.53	0.06–5.1	0.59
Sapien	1.3	0.33–5.6	0.68
Corevalve	0.82	0.2–3.4	0.78
**Baseline and Day 1 biological parameters**			
BNP			
Baseline	1	0.99–1	0.73
Day 1	1	1–1.1	0.28
Serum Creatinine			
Baseline	1	0.99–1	0.36
Day 1	1	0.99–1	0.1
Serum Cystatin			
Baseline	1.2	0.6–2.2	0.59
Day 1	1.6	1–2.5	0.05
**Baseline and Day 1 Echocardiography parameters**			
Baseline LVEF	0.97	0.93–1	0.29
Baseline Mean Aortic Gradient	0.99	0.94–1.1	0.72
Right Atrial Pressure (RAP)			
Baseline	1.1	1–1.3	0.035
Day 1	1.1	0.97–1.7	0.11
Stroke Volume			
Baseline	1	0.98–1.1	0.28
Day 1	1	0.95–1.1	0.77
Cardiac index			
Baseline	1.6	0.82–3.1	0.17
Day 1	1.1	0.51–2.3	0.86
**Renal doppler parameters**			
**High Renal Resistive Index (RRI > 0.8)**			
Baseline	0.9	0.18–4.8	0.935
Day 1	6.5	1.3–32.9	0.021
**Renal hemodynamic parameters**			
Renal pulse pressure			
Baseline	1	0.98–1.1	0.38
Day 1	0.98	0.93–1	0.38
Renal Arterial Load			
Baseline	1	0.85–1.2	0.87
Day 1	0.97	0.78–1.2	0.76
High Renal Arterial Compliance (> 0.12)			
Baseline	1.5	0.355–6.337	0.581
Day 1	1.325	0.315–5.565	0.701
**Systemic hemodynamic parameters**			
Valvulo-arterial impedance			
Baseline	0.93	0.56–1.5	0.77
Day 1	1.3	0.96–1.9	0.083
Total arterial load			
Baseline	1	0.53–1.9	0.95
Day 1	1.8	0.89–3.6	0.099
Pulse Pressure			
Baseline	1	0.99–1.1	0.26
Day 1	1	0.95–1	0.85
Systemic arterial compliance			
Baseline	0.67	0.51–8.7	0.76
Day 1	0.33	0.53–2	0.23
Resistive arterial load			
Baseline	1	0.99–1	0.67
Day 1	1	0.98–1.01	0.14

Data are presented as mean ± or *n* (%). ACE inhibitor = Angiotensin-converting enzyme inhibitor; AKI = Acute Kidney Injury; ARBs = Angiotensin II receptor blockers; BMI = body mass index; BNP = B-type natriuretic peptide; CI = confidence interval; Cr = creatinine; CRP = C-reactive protein; CysC = Cystatin; EuroSCORE = logistic EuroSCORE predicted risk of mortality at 30 days; HR = Hazard Ratio; LVEF = left ventricular ejection fraction; sCr = serum creatinine; sCyC = serum Cystatin C.

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
