# Peer review of "Bedside Renal Doppler Ultrasonography and Acute Kidney Injury after TAVR"

_jcm, 2020, doi:10.3390/jcm9040905_

Round 1

Reviewer 1 Report

As TAVR procedure is becoming a common mode of aortic valve replacement I would like to congratulate authors for conducting the study on the burning issue of acute kindey injury (AKI) complicating TAVR. 

The most prognostic factor that is baseline kidney function is frequent  among patients with severe aortic stenosis (AS) thus other risk factors are needed. The ideal tool should  be sensitive, specific and preceed the standard measures, that is diagnosis. Renal resistance index (RRI) as proposed by the authors is worth consideration.

I find the article well structured and with clear message.  I am impressed that ultrasound was performed by two independent physians which confirms objectivity of the study.

Despite that, I would like to address few questions/suggestion:

1)Line 77:

In the part entitled „ Definition of AkI” the authors present AKIN Classification that suggests that urine output criteria were used in the study. As far as I understand (based on Table 4) the classification used in the study is solely based on biochemical markers. If so please state it clearly.

2) What is more, in the RESULTS section there is no information on AKI stages rates.

3)Line  145

In Table 1 should be extended with periprocedural data such as type of analgesia ( light sedation vs total analgesia), presence periprocedural complications (cardiac arrest, inotropic support etc), RBC transfusion

4)Additionally, since RRI baseline was elevated in the whole cohort and the optimal RRI cutoff point at 1 day was 0.795 that is within the range of baseline RRI(0.76±0.7) It would be worth to assess if % change of RRI plays role in predicting AKI.

5) I would also advise stating in the limitations/results that AUC of 0.766 is of moderate prognostic and this might be related to low number of events. 

6) In my opinion two important outcomes are missing:

- renal replacement therapy

-persistent kidney injury. Was there a deterioration of chronic kidney diease at discharge? My suggestion would be to evaluate if RRI was a factor to indicate AKI patients at risk of chronic deterioration ( change in CKD class) . Due to low numer of cases that might be challenging but for sure worth mentioning.

7) Finally,

I do have objections regarding statistics. Since the authors stated that for non-parametric variabes Mann Whitney test was used these variables should be presented as median with IQR. It is questionable if any of variables in AKI group had a normal distribution since the numer of cases is very low.

As most of data in table 1 is insignificant I presume even when assessed by nonparametric tests are expected to remain insignificant.

The major concern regards RRI values. Please confirm its normality distribution and which test was used for the comparioson between AKI and nonAKI patients.

Author Response

REFEREE COMMENTS:

Referee: 1

Comments to the Author

As TAVR procedure is becoming a common mode of aortic valve replacement I would like to congratulate authors for conducting the study on the burning issue of acute kidney injury (AKI) complicating TAVR. 

The most prognostic factor that is baseline kidney function is frequent among patients with severe aortic stenosis (AS) thus other risk factors are needed. The ideal tool should be sensitive, specific and proceed the standard measures, that is diagnosis. Renal resistance index (RRI) as proposed by the authors is worth consideration. 

I find the article well-structured and with clear message.  I am impressed that ultrasound was performed by two independent physicians which confirms objectivity of the study.

Despite that, I would like to address few questions/suggestions:

Comments:

1)Line 77:

In the part entitled „ Definition of “AKI” the authors present AKIN Classification that suggests that urine output criteria were used in the study. As far as I understand (based on Table 4) the classification used in the study is solely based on biochemical markers. If so please state it clearly.

We would like to thank the Reviewer for his pertinent comment. The definition of acute kidney injury (AKI) in our study relied on both biochemical markers and urine output criteria. As AKI is an important clinical issue in the field of TAVR, we dedicated ourselves to define AKI according to current existing guidelines. All TAVR patients had urine output recorded in our cardiovascular intensive care unit (CICU) after the procedure. However, no patient showed AKI according to current and validated urine output criteria as stated in the AKIN classification system. As pointed out by the Reviewer and in order to leave a clear and understandable message to the JCM readers, this is now more clearly stated in the Methods section

Definition of AKI

Acute kidney injury (AKI) by serum Creatinine (sCr) (AKIsCr) was defined according to the VARC-2 definition (1) as an absolute increase in serum creatinine of ≥0.3 mg/dL
(≥26.4 μmol/L) OR ≥ 50% increase in serum creatinine 72 hours after TAVR. The AKIN Classification classified biochemical severity of AKI as follow: (i) Stage 1: Increase in serum creatinine to 150–199% (1.5-1.99×increase compared with baseline) OR Urine output 0.5 mL/kg/h for .6 but ,12 h (ii) Stage 2 : Increase in serum creatinine to 200–299% (2.0–2.99 × increase compared with baseline) OR Urine output ,0.5 mL/kg/h for .12 but ,24 h (iii) Stage 3 : Increase in serum creatinine to ≥300% (>3×increase compared with baseline) OR Urine output ,0.3 ml/kg/h for 24 h OR Anuria for 12 h.

2) What is more, in the RESULTS section there is no information on AKI stages rates. 

As requested by the reviewer, renal failure according to the AKIN classification is now more clearly stated in the results section

Results - Baseline Characteristics

Regarding AKI as determined by serum creatinine (AKIsCr), stage 1 AKI according to the AKIN system occurred in 8 patients, stage 2 in 2 patients and no stage 3.

3)Line 145

In Table 1 should be extended with periprocedural data such as type of analgesia (light sedation vs total analgesia), presence periprocedural complications (cardiac arrest, inotropic support etc.), RBC transfusion

All patients in our cohort underwent transcatheter aortic valve replacement under conscious sedation. No general anesthesia was indeed performed. No per-operative complications could be evidence such as cardiac arrest, tamponade, immediate red blood cell transfusion nor vasopressor and/or inotropic support required.

To take into account the Reviewer’s comment, two sentences concerning the procedural characteristics have been added.

Methods section - Patients

Commercially available valves, such as the balloon expandable Edwards SAPIEN XT or S3 prosthesis (Edwards Life sciences LLC, Irvine, CA) and the self-expandable CoreValve or Evolute-R (Medtronic CV, Irvine, CA), were used. Anesthesiological management included a local anaesthesia plus sedation. It consisted of lidocaine injected subcutaneously at the arterial and venous access sites (maximum dose 4 mg/kg), with sedation accomplished with remifentanil infusion adjusted according to the patient's response (target level: score 2-3 with modified Wilson sedation scale; starting dose 0.025 μg/kg/min, maximum dose 0.2 μg/kg/min).

Results section - Baseline Characteristics

No per-operative complications could be evidence such as cardiac arrest, tamponade, immediate red blood cell transfusion nor vasopressor/inotropic support requirement. Likewise, no patients required dialysis therapy.

4)Additionally, since RRI baseline was elevated in the whole cohort and the optimal RRI cutoff point at 1 day was 0.795 that is within the range of baseline RRI (0.76±0.7) It would be worth to assess if % change of RRI plays role in predicting AKI.

We thank the reviewer for this constructive remark. We performed a pilot study to test the association of renal resistive indices with AKI after TAVR. Therefore, we ‘only’ included 100 patients, prospectively, in whom 10 developed AKI and we reported a significant association of AKI and RRI at day 1.  As acknowledged by the reviewer, the sample size is small. Moreover with 10 events, any further analysis regarding % change in RRI in predicting AKI is going to be extremely underpowered, tentative and hypothesis generating. 

As a pilot study, we disclose in this manuscript proposed for publication in JCM: preliminary results and initial finding. We will dedicate ourselves to further extended work in the field of renal doppler assessment and TVAR with a bigger cohort. Also, such analysis proposed by the reviewer will be of particular interest.

5) I would also advise stating in the limitations/results that AUC of 0.766 is of moderate prognostic and this might be related to low number of events. 

Changes have been made according to the Reviewer’s request.

Study limitations Section

The AUC of 0.766 should be considered with caution and interpreted in the light of low number of events. 

6) In my opinion two important outcomes are missing:

6.1 Renal replacement therapy

6.2 Persistent kidney injury. Was there a deterioration of chronic kidney disease at discharge? My suggestion would be to evaluate if RRI was a factor to indicate AKI patients at risk of chronic deterioration (Change in CKD class). Due to low number of cases that might be challenging but for sure worth mentioning. 

6.1 In the present study, no patients required dialysis therapy. To address the reviewer’s concern, this is now more clearly stated in the Methods and results section

Methods section – Definition of AKI

Patients were excluded if they were receiving dialysis at baseline.

Results section - Baseline Characteristics

No per-operative complications could be evidence such as cardiac arrest, tamponade, immediate red blood cell transfusion nor vasopressor/inotropic support requirement. Likewise, no patients required dialysis therapy.

6.2 So far, limited data exists on the entire spectrum of renal variations after TAVR. While AKI has been extensively investigated, both (i) cardio-renal hemodynamics and (ii) unchanged or even improved renal function at discharge after TAVR (Azarbal A et al.  Am J Cardiol. 2018) (Nijenhuis VJ et al. Am J Cardiol. 2018) (Roberto J et al. J Am Coll Cardiol. 2018 Supplement) have just been lately described. Although the description of the mechanisms relating mid or long-term renal variations after TAVR (either altered, unchanged or improved) and cardio-renal hemodynamics were far beyond the scope of the present study. However, such renal physiopathological pathways should be considered in upcoming studies. As pointed out by the reviewer the small sample size together with the absence of mid et long term follow-up limit our ability to address this issue.

7) Finally,

I do have objections regarding statistics. Since the authors stated that for non-parametric variables Mann Whitney test was used these variables should be presented as median with IQR. It is questionable if any of variables in AKI group had a normal distribution since the number of cases is very low.

As most of data in table 1 is insignificant I presume even when assessed by nonparametric tests are expected to remain insignificant.

As requested by the Reviewer, the normality of each variable was tested in the whole cohort using Kolmogorov-Smirnov test. Normality was confirmed expect for Hb, stroke volume. Those data are presented as median (IQR) in the revised version of the manuscript.

It is now clearly stated in the statistical analysis section :

Normality of the distribution was tested using Kolmogorov-Smirnov Test

The major concern regards RRI values. Please confirm its normality distribution and which test was used for the comparison between AKI and nonAKI patients.

A normal distribution could be evidenced for RRI values by Kolmogorov-Smirnov test. Comparison between AKI and non AKI patients were performed using ANOVA tests. 

Reviewer 2 Report

- Rational of the study in unclear and should be reported. “To date there are no data regarding the beneficial use of intrarenal hemodynamic and Doppler based resistive index (RRI) to predict AKI and outcomes in a large cohort of TAVR patients” Authors are not reporting any beneficial data of RRI, they are just investigating a possible association between RRI and AKI development. The possible clinical effects of RRI application are not studied.

- “100 patients with severe AS and high or intermediate surgical risk according to Logistic EuroSCORE”. Please report a reference

- Power calculation is lacking

- There is no information about the ROC analysis in the methods section (statistical analyses)

- Results presentation requires an extensive revision. The amount of data reported is excessive. It should be shortened and focused only on a limited number of results (only relevant results concerning the question underlying the study should be reported). Overall the data presentation is confusing and difficult to read. On the contrary, results of multivariate analyses should be reported in a table (they are impossible to be evaluated without reporting all variables considered). Please, edit the discussion accordingly

- Authors should report in the tables legend the statistical analysis that was used

  • The number of abbreviations used is very high and this has a negative impact on the reader. In the abstract, explanation of AKI abbreviation in repeated twice.

Author Response

Referee: 2

Comments to the Author

  1. Rational of the study in unclear and should be reported. “To date there are no data regarding the beneficial use of intrarenal hemodynamic and Doppler based resistive index (RRI) to predict AKI and outcomes in a large cohort of TAVR patients” Authors are not reporting any beneficial data of RRI, they are just investigating a possible association between RRI and AKI development. The possible clinical effects of RRI application are not studied.

To accommodate the Reviewer comment we have softened our statement

To date there are little data regarding cardio-renal hemodynamics in the field of TAVR. Therefore, the aim of this pilot study was to (i) investigate the association of Doppler based RRI with acute kidney injury after the TAVR procedure and (ii) evaluate the dynamic change of various cardio-renal hemodynamic parameters according to AKI occurrence.

  1. “100 patients with severe AS and high or intermediate surgical risk according to Logistic EuroSCORE”. Please report a reference

A reference has been added in the revised manuscript as requested.

Table 1. Baseline characteristics

Euroscore*

*The logistic European System for Cardiac Operative Risk Evaluation (EuroSCORE) is calculated by means of a logistic-regression equation; online and downloadable versions of the EuroSCORE calculator are available on the EuroSCORE Web site

  1. Power calculation is lacking

We would like to thank the Reviewer for his pertinent comment. As stated in the introduction section, we only sought to conduct a pilot study. Following current reference and recommendations regarding pilot studies (Moore CG et al.  Recommendations for planning pilot studies in clinical and translational research. Clin Transl Sci. 2011) we did not perform power calculation. Indeed, we did not test a hypothesis, but focused on cardiorenal interactions – AKI and RRI measurements.

We fully agree with the reviewer concerns and are aware that many journal comities recommend power calculation to be performed for all studies, even pilot ones; but we did not want to feel compelled to include hypothesis tests that are not the real reason for conducting our study.

  1. There is no information about the ROC analysis in the methods section (statistical analyses)

As requested by the reviewer, ROC analysis is now clearly reported in the methods section (statistical analysis)

Methods - Statistical analysis

Receiver-operating characteristic (ROC) curve analysis was performed to establish the threshold values most predictive of AKI

  1. Results presentation requires an extensive revision. The amount of data reported is excessive. It should be shortened and focused only on a limited number of results (only relevant results concerning the question underlying the study should be reported). Overall the data presentation is confusing and difficult to read. On the contrary, results of multivariate analyses should be reported in a table (they are impossible to be evaluated without reporting all variables considered). Please, edit the discussion accordingly

To take into account the reviewer’s concern, the paragraph “Prognostic impact of AKIsCyC” was deleted; CRP values which are of little incremental values (table 3 Biological parameters) have been deleted. As requested, the results of multivariate analysis are now clearly reported in a specific table (table 9). We thank the reviewer for his comments as the revisited manuscript has gained from clarity and now only focusing on RRI – systemic and renal hemodynamics – and AKI.

  1. Authors should report in the tables legend the statistical analysis that was used

Comparison between variables were performed using ANOVA tests except for Hb and Stroke Volume that were tested using Mann-Whitney.

  1. The number of abbreviations used is very high and this has a negative impact on the reader. In the abstract, explanation of AKI abbreviation in repeated twice.

The paper has been fully rewritten regarding the use of abbreviations. Changes have been made according to the Reviewer’s request in the abstract section.